# Cross-device federated unsupervised learning for the detection of anomalies in single-lead electrocardiogram signals

**Maximilian Kapsecker**[1,2]*, **Stephan M. Jonas**[2]

**1** TUM School of Computation, Information and Technology, Technical University of Munich, Garching bei München, Bavaria, Germany, **2** Institute for Digital Medicine, University Hospital Bonn, Bonn, North Rhine-Westphalia, Germany

* max.kapsecker@tum.de

**Data availability statement:** The data set for this study is publicly accessible via the following reference: Icentia11k database (https://www.physionet.org/content/icentia11k-continuous-ecg/1.0/).

**Funding:** The author(s) received funding from the Bavarian State Ministry of Health and Care as part of the DigiMed Bayern research project.

**Competing interests:** The authors have declared that no competing interests exist.

## Abstract

Background: Federated unsupervised learning offers a promising approach to leveraging decentralized data stored on consumer devices, addressing concerns about privacy and lack of annotation. Single-lead electrocardiograms (ECGs) captured on consumer devices are of particular interest due to the global prevalence of cardiovascular disease. The combination of federated and unsupervised learning on biomedical data in a cross-device environment raises questions regarding feasibility and accuracy, especially when considering heterogeneous data. Methods: A randomly selected subset of the Icentia11k open-source dataset containing mobile ECG recordings was used for this study. Heartbeats are labeled as normal, unknown or the pathological classes: premature atrial contraction and premature ventricular contraction. A linear autoencoder model was used as a method to predict the pathological cases using the embedding space and reconstruction error. The model was integrated into a mobile application that supports ECG data recording, preprocessing into heartbeat segments, and participation in a federated learning pipeline as a client node. The autoencoder was trained collaboratively using federated learning with twenty mobile devices, followed by an additional ten epochs of on-device fine-tuning to account for personalization. Results: The approach yielded a sensitivity of 0.87 and a specificity of 0.8 when the predicted anomalies were compared with the ground truth in a binary fashion. Specifically, the detection rate for premature ventricular contraction was excellent with a sensitivity of 0.97. Conclusion: Overall, the approach proved to be feasible in implementation and competitive in accuracy, specifically when the model was fine-tuned to the subject's data.

## Author summary

Advances in consumer devices have enabled the recording of electrocardiogram (ECG) signals independent of time and location, representing a significant step forward in

the management of cardiovascular diseases. However, implementing anomaly detection in such decentralized environments presents several challenges, including privacy concerns, the unsupervised nature of self-recording, and the heterogeneity of data. To address these challenges, we explored the use of federated learning with autoencoder models in a mobile environment for single-lead ECGs. Our approach was tested in a real-world setting using an open-source wearable ECG dataset, supported by the development of a mobile application for data management and collaborative model optimization. To mitigate the impact of data heterogeneity, we implemented on-device fine-tuning of the federated base model for personalization. The anomaly detection process, utilizing the distribution of the embedding space and reconstruction error, demonstrated high sensitivity, although specificity was slightly compromised, likely due to the presence of noisy samples. Future improvements, such as automatic artifact removal and enhanced capabilities for unobtrusive data processing and training, are expected to advance the application toward widespread end-user adoption. Overall, our findings support the potential of federated anomaly detection and highlight promising directions for future mobile health applications of similar characteristics.

## Introduction

### Overview and motivation

In recent years, advancements in mobile device hardware and software have transformed smartphones and tablets from simple communication tools into powerful computational platforms equipped with advanced sensors. These devices are now capable of recording, processing, and analyzing high-resolution biomedical signals directly on-device [1]. The electrocardiogram (ECG) is particularly relevant as a diagnostic tool for cardiovascular diseases, which are the leading cause of global mortality [2]. Consumer devices such as the Polar H10 allow long-term ECG recording beyond clinical settings, independent of time and location [3]. Although a standard clinical ECG comprises 6 to 12 leads, a single-lead ECG can still show pathological patterns [4].

Performing computations directly on consumer devices, such as smartphones, offers data privacy and reduced communication since data does not need to be transferred to a shared processing node. In particular, on-device machine learning can help with complex tasks typically performed by professionals, such as detecting anomalies in the ECG, identifying baseline changes, and classifying the findings.

However, training machine learning models on the device using local data poses challenges. Learning from a single subject carries the risk of overfitting and poor generalization. Deep learning methods such as neural networks might capture intra-person variations but can fail to distinguish between regular and suspicious cardiac activity due to the lack of comparative data. For instance, learning from the ECG of a person with heart disease can cause the model to inaccurately consider these signals as normal without looking at a broader, healthier population. Furthermore, in mobile environments, annotations of healthy beats or anomalies are not available by default, limiting machine learning efforts to unsupervised methods. The challenges identified are threefold: (1) the need for privacy-preserving methods [5–7], (2) lack of annotations [8], and (3) recordings of ECGs by different subjects are non-independent and identically distributed (non-IID) and may suffer from noise induced by the mobile environment [9,10].

**Table 1. Studies using FL for the detection of ECG anomalies.**

| Study | Datasets[*] | Target Client | Clients | Setting |
|---|---|---|---|---|
| *Goto et al. (2022)* [11] | Self-collected from four hospitals | Institutions | 4 | Supervised |
| *Sakib et al. (2021)* [12] | MIT-BIH Supraventricular Arrhythmia [13,14], MIT-BIH Arrhythmia [14,15], INCART 12-lead Arrhythmia [14], Sudden Cardiac Death Holter [14,16] | Ultra-Edge Nodes | 2-10 | Supervised |
| *Asif et al. (2023)* [17] | MIT-BIH Arrhythmia [14,15] | Institutions | 4 | Supervised |
| *Lee et al. (2020)* [18] | PhysioNet/CinC Challenge ECG 2017 [14,19] | Institutions | 3 | Supervised |
| *Weimann et al. (2024)* [20] | PhysioNet/CinC Challenge ECG 2021 [14,21,22], PTB-XL [14,23,24], Ningbo and Chapman-Shaoxing [14,25,26] | Institutions | 5 | Supervised |
| *Yuan et al. (2020)* [27] | PhysioNet/CinC Challenge ECG 2017 [14,19] | IoT Devices | 16-64 | Supervised |
| *Zhang et al. (2020)* [28] | Not available | Institutions | 100 | Supervised |
| *Tang et al. (2021)* [29] | Self-collected from eight hospitals | Institutions | 8 | Supervised |
| *Lin et al. (2022)* [30] | MIT-BIH Arrhythmia [14,15] | IoT Devices | 24 | Supervised |
| *Sun et al. (2022)* [31] | MIT-BIH Arrhythmia [14,15], MIT-BIH Supraventricular Arrhythmia [13,14], INCART 12-lead Arrhythmia [14] | IoT Devices | - | Supervised |
| *Raza et al. (2022)* [32] | MIT-BIH Arrhythmia [14,15] | Raspberry Pi | 3 | Supervised |
| *Chorney et al. (2024)* [33] | MIT-BIH Arrhythmia [14,15], PhysioNet/CinC Challenge ECG 2017 [14,19], PTB-XL [24] | Institutions | 7 | Supervised |
| *Jimenez et al. (2023)* [34] | PhysioNet/CinC Challenge ECG 2020 [14,35] | Edge Devices | 2-10 | Supervised |
| *Ying et al. (2023)* [36] | MIT-BIH Arrhythmia [14,15] | Simulated IoT Devices | 100 | Semi-supervised |
| *Raza et al. (2023)* [37] | BIDMC Congestive Heart Failure [14,38] and MIT-BIH Normal Sinus Rhythm [14] | Raspberry Pi | 3-5 | Unsupervised |
| *Rajagopal et al. (2023)* [39] | ECG5000 [14] | Simulated Edge Devices | - | Unsupervised |
| This work | Icentia11k [14,40,41] | Smartphones | 20 | Unsupervised |

[*] Unspecified datasets or those with a missing or incorrect reference were not listed.

As part of this proof-of-concept study, a publicly available, mobile-recorded ECG data set was used and the data transferred to smartphones. A custom developed smartphone application enabled privacy-preserving and unsupervised learning routines on this data for the detection of anomalies. In addition, the implementation supported personalization by allowing fine-tuning of a federated trained model on local data, thereby accounting for non-IID effects.

## Related work

Since its inception in 2017, federated learning (FL) has emerged as a key technology for privacy-preserving and decentralized machine learning, enabling collaborative model training without compromising individual privacy [42]. FL has been studied and applied predominantly in the context of supervised learning [42], including for healthcare purposes [43,44] such as ECG prediction (cf. Table 1). In particular, FL was utilized for the detection of hypertrophic cardiomyopathy [11] and cardiac arrhythmia [12,17,18,20,27,34]. The latter use case was also extended to investigate non-IID ECG data and its application to personalization techniques [28,29]. Solutions such as *FedCluster* were introduced to overcome the challenges of non-IID ECG data [30]. *FedCluster* weights and clusters clients' updates based on similarity, which improves the model's ability to diagnose rare conditions by facilitating knowledge transfer between similar clients. The framework around *FedCluster* can also handle highly unbalanced ECG distributions by sharing ECGs from underrepresented classes from a globally available dataset. Compared to the traditional *FedAvg* approach [45], the accuracy on the MIT-BIH dataset increased by up to 52% for clients with

highly skewed data distributions, while the overall accuracy improved slightly from 89.09% to 89.26%.

*Sun et al. (2022)* introduced SCALT, an FL framework that first extracts features from a signal using 1-D convolutional operations and uses a dedicated mini-classifier for each class [31]. As a result, SCALT ensures scalability and transferability to different tasks and achieved an arrhythmia classification accuracy of 98.65%, 98.29%, and 98.05% on the MIT-BIH, MIT-BIH supraventricular, and INCART 12-lead arrhythmia datasets, respectively. *Raza et al. (2022)* used an autoencoder for signal denoising and then incorporated the signals for arrhythmia detection in a federated setting [32]. The accuracy, evaluated using 5-fold cross-validation, was up to 98.9% based on the denoised MIT-BIH dataset. Similarly, *Chorney et al. (2024)* employed an autoencoder for signal denoising as input to FL [33]. The focus was on common challenges and complexities in healthcare, including inconsistent input dimensions, varying features in the data, and data distributed across multiple facilities without permission to share. The method achieved an accuracy, precision, recall, and F1 score of 73.0%, 66.6%, 73.0%, and 69.7%, respectively, in the prediction of cardiac arrhythmias.

Instead of using autoencoders for signal denoising, autoencoders can also be used to directly detect anomalies in an unsupervised manner while fitting well into the investigated supervised FL environment due to their self-supervision. The objective of this work is to use FL with autoencoder models to learn a representation of ECG signals from decentralized mobile devices and detect anomalies. Previous research has already explored the possibilities of embedding unsupervised and semi-supervised ECG anomaly detection in FL technologies.

*Ying et al. (2023)* introduced the *FedECG* framework for semi-supervised FL on unlabeled ECG data, which improved privacy with only minimal reduction in accuracy [36]. *FedECG* was motivated by similar challenges addressed in the underlying work, including the need for privacy, the lack of annotations, and non-IID data. *FedECG* included signal smoothing, segmentation of individual heart beats, and conversion of these segments into images using the Gramian Angular Fields technique. A lightweight version of ResNet-9 was trained on a batch of annotated data on the server-side and injected on the client-side to predict pseudo-labels based on weakly augmented data. Predictions were then made on the same but heavily augmented data to compute the loss relative to the pseudo-labels, which yielded the loss for model optimization. To solve the problem of non-IID data, which is generally problematic for the *FedAvg* algorithm, subjects were clustered by model weights using the k-means algorithm. The experimental results on the MIT-BIH dataset (half of the data was labeled) showed an accuracy of 94.8%, which is about 2% lower than centralized state-of-the-art methods, but has the advantage of enhanced privacy.

*Raza et al. (2023)* used (variational) autoencoders with transformer layers in an FL pipeline to detect anomalies [37]. The approach, called *AnoFed*, identified anomalies based on reconstruction loss, which tends to be higher for anomalous data because the model has rarely learned from these samples. The authors integrated a support vector machine into their FL framework to learn the best reconstruction error threshold for distinguishing between healthy and anomalous samples. *AnoFed* showed accuracies between 93% and 98.8%. Similarly, *Rajagopal et al. (2023)* used the strategy of calculating the reconstruction error of a collaboratively trained autoencoder as a predictor of anomalies in ECG signals in an *Internet of Things* (IoT) environment [39]. Although the focus of the work was on evaluating energy consumption, network communication, and execution time, the accuracy in detecting anomalous samples was about 95%.

### Research questions

The gap identified in previous research is primarily the dependence on clinically recorded ECGs (e.g., the MIT-BIH dataset), the lack of application in a consumer device setting, and few investigations into federated unsupervised learning (cf. Table 1). Although unsupervised ECG anomaly detection was successfully applied in previous studies, its implementation in the emerging domain of wearable devices remains underexplored [46]. In addition to the challenges posed by the integration of unsupervised learning with resource-constrained wearable devices, the complexity of non-IID data and data privacy requirements must be considered. Federated learning has proven to be effective for supervised tasks as a means of preserving privacy [42]; however, to the best of the authors' knowledge, few studies have investigated its application in an unsupervised setting for ECG anomaly detection. Moreover, these few studies were performed in simulated environments [36,39] or on non-consumer devices [37], limiting their generalizability to real-world outpatient scenarios. Therefore, this work focuses on the implementation of a real-world FL scenario. An electrocardiogram dataset recorded in a mobile setting is used and autoencoders are employed for anomaly detection. A fine-tuning approach is adopted to handle non-IID data. The research questions (RQ) derived from the motivation and the related work are as follows:

**RQ1**  Feasibility: How feasible is autoencoders implementation for federated unsupervised learning, in a cross-device real-world scenario?

**RQ2**  Anomaly Detection: What is the performance of autoencoders in the detection of electrocardiogram anomalies as part of a cross-device federated unsupervised environment?

**RQ3**  Inter-subject variations: Can on-device fine-tuning be sufficient for personalization and detecting baseline changes, thus addressing the challenge of non-IID data?

## Material and methods

To conduct experiments on federated unsupervised learning in a cross-device environment, an annotated dataset for evaluation purposes, a machine learning model that can be executed on mobile hardware, and an associated application for conducting studies were required. The materials and methods are supported by an implementation that is available at https://github.com/CardioKit/cardioflow.

### Data

Icentia11k is an open-source dataset of 11,000 participants' ECG recordings [14,40,41]. The dataset was primarily collected from 2017 to 2018 in Ontario, Canada, comprising 45.3% female, 42.6% male, and 12.2% participants of unknown sex. The age distribution approximates a bell-shaped curve, with a mean age of 62.2 years and a standard deviation of 17.4 years. Although there is an average inclination towards an older population, the ages range widely from adolescents to individuals close to 100 years old. The long-term recordings consist of single-lead ECG signals collected over 3 to 14 days and sampled at 250 Hz. Approximately 2.7 billion beats were categorized as either normal, premature atrial contraction (PAC), premature ventricular contraction (PVC), or undefined, i.e., unclassifiable beat (cf. Fig 1 and S1 Fig). The data are particularly well suited for this study because the recordings were made in a mobile environment and annotations are available at the beat level for evaluation purposes.

A subset of twenty subjects was randomly selected from Icentia11k, with the number of subjects determined by the availability of mobile devices. The data for each subject consist of

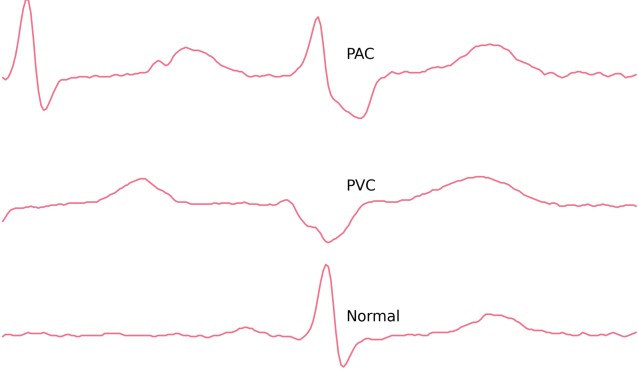

**Fig 1. Single heartbeats with annotations.**

fifty segments, each representing approximately 70 minutes of ECG recording, resulting in a substantial total recording time. The segments were parsed into a comma-separated values file, with the signal values and annotations arranged in columns. The annotation "+" indicating a rhythm label was ignored for later analysis and treated as no annotation. Although arrhythmia can alter the morphology of single-lead ECG heartbeats, accurate identification typically requires analyzing consecutive heartbeats. Therefore, using arrhythmia annotations based solely on single beats may result in medically unreliable samples. *Normal*, *PAC*, and *PVC* were the only relevant annotations for analyzing individual heartbeats. These annotations were available approximately at the R-peak positions.

## Autoencoder for anomaly detection

The architecture of an autoencoder consists of two stacked neural networks, the encoder $E_\phi$ and the decoder $D_\theta$, parameterized by the weights $\phi$ and $\theta$, respectively. The encoder maps data from a high-dimensional space $V \subset \mathbb{R}^n$ into an embedding space $L \subset \mathbb{R}^k$ of low dimension, where $k \ll n$, formally expressed as $E_\phi : V \to L$. The low-dimensional representations $l \in L$ are referred to as the embedding vectors. In a standard setting, the decoder $D_\theta : L \to \mathbb{R}^n$ attempts to reconstruct the original input from the embedding vector. Accurate reconstruction is achieved if the $k$ latent dimensions effectively encode the predominant signal information. Thereby, the autoencoder is optimized by adjusting the weights $\phi$ and $\theta$ to minimize the reconstruction error through an appropriate loss function.

For this study, the autoencoder model consists of a simple multi-layer perceptron architecture, with the encoder and decoder consisting of two dense layers (cf. Fig 2). The activation functions are the rectified linear unit (ReLU) for the inner dense layers and sigmoid for the output layer. The decision not to use more complex layer structures or a variational autoencoder that ensures a regularized latent vector space was primarily due to the limitations of the mobile machine learning framework CoreML (https://developer.apple.com/documentation/coreml/, retrieved on 24 November 2024). The dimension of the latent vector space was set to $k = 12$, based on the performance demonstrated in prior research by *Kapsecker et al. (2024)* [47]. Their study highlighted the effectiveness of a 12-dimensional latent space in embedding single-lead heartbeats using the autoencoder paradigm.

In a mobile environment, the model is usually in a compiled version that does not allow the results of intermediate layers to be evaluated. Therefore, the latent vector space was

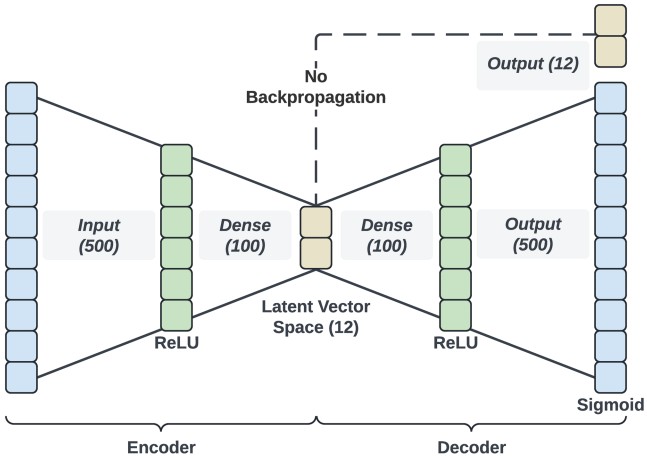

**Fig 2. Autoencoder model.**

defined as the output, but not used as the source for backpropagation (cf. Fig 2). The autoencoder architecture was developed using PyTorch and subsequently converted to CoreML format with *coremltools* (https://apple.github.io/coremltools/, retrieved on 24 November 2024).

A well-optimized autoencoder with strong generalization capabilities can detect anomalies when the ECGs are mostly normal. Specifically, an autoencoder that was trained primarily on healthy ECG data can accurately reconstruct the regular non-pathological patterns. When the model encounters rare pathological ECGs, the reconstruction error is significantly higher because it is less familiar with these anomalies. This difference in reconstruction error makes it easier to identify anomalous ECGs, such as pathological conditions.

Likewise, the embedding space can serve as a basis for the detection of anomalies. It is expected that most ECGs fall into a nearby region due to the similarity and prevalence of healthy samples. Anomalies and irregular ECG patterns tend to deviate from this central cluster and can be detected by setting a threshold for the distance to the center of the cluster.

## Mobile application

A prototype of a smartphone application called *Cardioflow* (https://github.com/CardioKit/cardioflow/tree/main/src/application, retrieved on 17 November 2024) was developed to consider a realistic cross-device FL setting. The target platform was iOS, mainly using the Swift implementation of the *Flower* framework [48,49], which provides a library for federated learning routines. Because iOS devices maintain a relatively homogeneous environment across different device generations and hardware capacities, this platform serves as a suitable basis for the prototype implementation. Since Flower also provides an Android implementation, extending the approach to a mixed-device setup remains a feasible direction for future work. In a first step, the functional requirements (FR) were defined:

FR1 The application must be able to receive an ECG signal – in a common format such as a comma-separated values file – including the option to provide annotations for visualization purposes. The collected data must be stored in an application-internal memory.

FR2 Processing the raw input signal as needed, including R-peak detection, smoothing, segmentation, normalization, upsampling, and signal quality assessment.

**FR3** Run machine learning on the device using an autoencoder model for the ECG data in order to fine-tune the model to the data from the device.

**FR4** Ability to participate in an FL pipeline as a client node using the device-internal machine learning mechanism of **FR3**.

**FR5** Detection of anomalous ECGs based on the embedding space and the reconstruction error of the autoencoder.

**FR6** Visualization of data and statistics, such as the preprocessed ECGs, the number of beats and anomalies, the reconstruction quality and loss for individual ECGs, and the embedding space of the autoencoder.

The non-functional requirements included the robust and fast execution of data processing and machine learning routines. The application largely follows a Model-View-ViewModel (MVVM) architecture pattern.

The **entry view** is the initial screen that displays the results of the various processing components. These include the total number of recognized R-peaks (beats), visualizations of slices through the embedding space, and the distribution of reconstruction errors (cf. Fig 3: Client 1).

The **ECG view** allows users to record and manage electrocardiogram signals and gain insights from the analysis. In the present implementation, ECG data is uploaded as a CSV file, which can also contain predefined annotations. Looking ahead to future work involving a patient-based study, it is intended to record an ECG using the Polar H10 device as a data source. After uploading, the ECG is displayed in a list below, showing the source and timestamp of the data addition (cf. Fig 3: Client 2). Navigating within the ECG element allows the user to see one-second segments of the ECG with centered R-peaks. In addition, the view offers the quality assessment of the ECG signal, the diagnosis with the possibility to modify it, and the mean residual value, which represents the average difference between the original and the reconstructed signal (cf. Fig 3: Client 3).

The **settings view** serves as a control center for almost all data-related operations (cf. Fig 3: Client 20). While most of the functions in this view typically run as background tasks, these functions can be triggered manually by the user for the sake of transparency and controlled execution. The segment ECG functionality uses the *PeakSwift* package [50] to preprocess the signal as follows: (1) Detect R-peaks using the Neurokit method presented in *Makowski et al. (2021)* [51]; (2) Smooth the signal (Butterworth filter of order 5 and low cut frequency of 0.5 followed by powerline filtering) provided as a by-product of Neurokit's R-peak detection; (3) Segment the signal into one-second windows around the R-peak (i.e., 500 milliseconds or 125 samples to each side) to extract single heartbeats, which are the subject of this study; (4) Upsample the segmented signals to a length of 500 by linear interpolation; (5) Map the values of the upsampled signals into the $[0, 1]$ range using min-max normalization; (6) Assign to each ECG trace a quality score by applying the method of *Zhao et al. (2018)* and following their class definitions and terminologies: *Excellent*, *Barely acceptable*, and *Unacceptable* [52]. If annotations are present, the signal is cut into one-second segments around the annotation point, otherwise around the detected R-peaks. The ECG prediction function triggers a pass through the autoencoder and stores the respective results in *CoreData*.

Machine learning routines can be triggered both within a device and in a federated setting. Participating as a client node in an FL environment using the *Flower* framework requires the input of the IP address and port of the orchestrating server [49]. For subsequent on-device model fine-tuning, two arguments can be adjusted to parameterize the local model optimization: the number of epochs and the batch size. Additionally, a static threshold can be set to compute the outliers from the embedding space and the reconstruction error, as outlined in

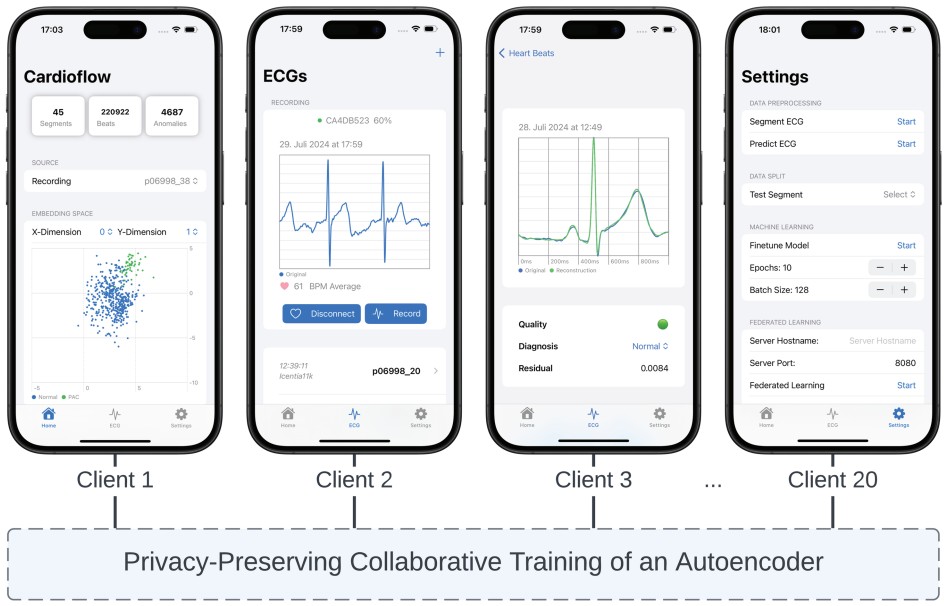

**Fig 3. The main views of the Cardioflow application, including the home screen (Client 1) displaying general statistics and analysis visualizations, the ECG recording and uploading interface (Client 2), the heartbeat reconstruction metrics (Client 3) showing model performance, and the settings panel (Client 20) for managing data processing and federated learning routines.**

Sect 2.2. Note that these parameters are typically not actively set by the client device but are instead computed, predefined as static properties, or provided by a central server. However, to explore different parameterizations and their associated behaviors within this study, we have included the ability to adjust these parameters on the fly.

## Technical setting

Cardioflow was deployed on twenty mobile devices running iOS version 17.5.1: 8 × iPhone 14 Pro, 7 × iPhone 13 Pro, 1 × iPhone 12 Pro, 2 x iPhone 11 Pro, 1 x iPad Pro (3rd generation), 1 x iPad Air (4th generation). The CSV files containing the electrocardiogram signals and annotations were transferred to the devices and stored in the application's document folder. The annotations were required solely for demonstration purposes, such as the visualization of clustering behavior illustrated in Fig 3 Client 1. The uploaded ECGs were preprocessed and segmented according to the described functionality in Sect 2.3 by clicking on *Start Segment ECG* in the settings tab – see S2 Fig for the experiment setting.

The FL orchestrating server was started on a MacBook Air M2 with 8 GB of memory, using the averaging strategy for weight aggregation (cf. S2 Fig). The arguments passed to the server included the minimum number of clients required, which was set to 20, and the number of optimization rounds, which was set to 250. Then, all devices were added to the pool of client nodes one by one by clicking on the corresponding *Start Federated Learning* button. Once the last device had joined the FL pipeline, the process started and the aggregated weights were stored on the MacBook running the FL server after each round.

During each round, five optimization epochs took place on the device. The test set was set to the first segment per subject, while the remaining segments were used for training. As the dataset creators randomly selected fifty segments from a larger pool, the first segment is

unlikely to represent the participants' initial recording. This approach ensures that the test data are free from bias due to initial learning effects and device adjustments. Loss was calculated using mean squared error and recorded after each round. With concluding the federated learning process, all devices shared the same model, which was referred to as the base model.

To facilitate personalization, the base model was independently fine-tuned on each device for ten epochs using a batch size of 128. This personalization step is hypothesized to mitigate data heterogeneity and enhance accuracy by adapting to local data. To prevent overfitting to local data distribution and preserve the model's generalizability – derived from training on a diverse range of participants – the duration of fine-tuning was carefully limited.

The same data, training and test distributions, and parameter settings were used to optimize the same model centrally on a single computer. The aim was to compare the centralized and federated models to determine the potential accuracy reduction caused by the federated setting. The centralized training was conducted on a Linux-based operating system with an *NVIDIA GeForce RTX 4090* for accelerated computing.

## Evaluation

The residuals, embeddings, annotations, and quality scores of the ECGs from the twenty devices were exported and transferred to a computer for analysis. The export was triggered after the execution of FL and once after the execution of the fine-tuning. The average residuals per segment were calculated to see the balance between the errors of the training and test segments and to detect possible overfitting. The t-distributed stochastic neighbor (t-SNE) method was used to reduce the data to a two-dimensional space and to visually explore the embedding space based on metadata such as annotations, subjects, and quality.

The residuals and embedded vectors were used as predictors for anomalies by determining a threshold that distinguishes normal from anomalous cases. Therefore, the embedded vectors were used on a per subject basis to calculate the Euclidean distance to their mass centroid, where a larger distance can indicate a subject-specific baseline drift. Formally, the anomaly score $a_k$ of the embedding vector $v_k$ for the $i$–$th$ subject is calculated as follows

$$a_k^{(i)} = d(v_k, \frac{1}{n} \sum_{j=1}^{n} v_j^{(i)})$$

where $d$ is the Euclidean distance and $n$ the total number embedded vectors of subject $i$. The formula implies that the center of mass is equal to the mean location of the embedding vectors $v_k$, which was assumed for this study.

The following computations were performed for all three results (i.e., residuals and embedding spaces) of the trained models, arising from either the FL routine (Fed), the fine-tuning process (FT), or the centralized (Cen) model optimization. The residuals and Euclidean distances were standardized for each subject to ensure that they are on the same scale so that they can be summed to a single value representing an anomaly score. Furthermore, the standardized values allow the comparison of anomaly scores between subjects. The area under curve (AUC) of the receiver operating characteristic (ROC) curve for distinguishing between normal and pathological cases was calculated using either residuals (Res) alone or the summation of residuals and embedding scores (ResEmb).

The DeLong test [53] was employed to determine whether the AUCs differ significantly, testing the alternative hypothesis that the true difference in AUC is not zero. In this context, the 95 % confidence interval (CI) for each AUC was computed. The optimal threshold was determined using the distance-based method for each individual based on their

respective ROC curves (see S4 Fig), as well as for the entire population. The latter is a generalized approach in which a single threshold is designed to fit all subjects. The integration of annotations in determining the best threshold does not adhere to a strictly unsupervised approach. However, this procedure demonstrates the discriminative power to separate normal from anomalous samples. In the mobile application, the most likely anomalous samples are ranked based on anomaly scores. Table 2 provides an overview of the models used for the evaluation assessments.

Finally, using the best thresholds, the confusion matrices and the associated metrics of specificity, sensitivity, and balanced accuracy values were computed for all subjects to evaluate performance. Due to the signal noise caused by the mobile environment, a high number of false positive predictions was expected. In combination with the imbalance of the classes, this was likely to lead to low F1 values, which supported the decision to use the balanced accuracy as an evaluation metric. To assess the impact of signal quality, the post-training analysis was repeated excluding signals classified as *barely acceptable* [52] and lower. The results of the subject-based analysis as well as insights into the reconstruction capabilities of the federated model were included in the supplementary material.

## Results

After deploying the long-term recordings of twenty subjects to the mobile devices, a total of $3,717,142$ ECG heartbeat segments were extracted on the devices from these recordings. Of these, $59,460$ samples represented anomalous ECGs, either PAC (n=$41,770$) or PVC (n=$17,690$). The signal quality of $1,377,102$ samples was categorized as *barely acceptable*, while $2,340,040$ samples were assigned *excellent* quality [52]. The correlation between the quality and the annotations was not significant with a Pearson correlation coefficient of 0.0071. All devices had at least one anomalous heartbeat, but the class distribution varied considerably. S3 Fig contains information on class distributions.

### Feasibility

All functional requirements were successfully implemented and tested as part of the study. The application allows users to upload ECG signals in comma-separated file format, with a size limitation to prevent freezing; files larger than approximately 200 MB may cause the app

**Table 2. Autoencoder model variants for the evaluation of the ECG anomaly detection.**

| Identifier | Setting | Description |
|---|---|---|
| Fed Res | Federated | Base autoencoder trained on data distributed to 20 devices. Anomalies identified using reconstruction residuals. |
| Fed ResEmb | Federated | Base autoencoder trained on data distributed to 20 devices. Anomalies identified using residuals and embedding vectors. |
| FT Res | Fine-tuned | Fine-tuned the base model locally on each device to adapt to individual data distributions. Anomalies inferred using residuals. |
| FT ResEmb | Fine-tuned | Fine-tuned the base model locally on each device. Anomalies inferred using residuals and embedding vectors. |
| Cen Res | Centralized | Autoencoder trained on centrally stored data. Anomalies identified using residuals. |
| Cen ResEmb | Centralized | Autoencoder trained on centrally stored data. Anomalies identified using residuals and embedding vectors. |

to freeze due to limited RAM availability and requires batching. The preprocessing of the signals and the execution of the machine learning on the device ran smoothly and without limitations, such as blocking the user interface (UI). FL also performed without complications and there were no significant performance degradations due to communication bottlenecks. However, the authors acknowledge that in a real-world setting, more devices are likely to participate, especially in a cross-device scenario.

One limitation of iOS is the lack of permission to execute background tasks in a controlled manner, e.g., at a specific time or for a specific duration, which limits the feasibility of FL. Nevertheless, tasks that do not require on-demand availability, such as the segmentation of recorded data, can be executed in the background.

Calculating outliers based on ECG signal reconstruction and embedding space was straightforward. The integrated SwiftUI (https://developer.apple.com/xcode/swiftui/, retrieved on 18 November 2024) library has a declarative style, which allowed for effective visualization of results and metrics, but further consideration is needed when presenting to non-expert users, such as in the case of a clinical trial.

## Anomaly detection

Table 3 compiles the results, which are divided into two approaches: (1) determination of an overall threshold for all samples (cf. Sect 3.2.1) and (2) individual determination of the best threshold per subject (cf. Sect 3.2.2). The results for the individual cases were only presented for anomaly detection based on the residual and embedding score of the fine-tuned model (FT ResEmb), as this approach generally performed best.

**Generalized approach**   Using the reconstruction error in combination with the distance to the center of mass (ResEmb) as a predictor for anomalies resulted in the best AUC values. The corresponding AUC are 0.816 (95 % CI: [0.8148, 0.8176]) for the federated model, 0.872 (95 % CI: [0.8708, 0.8735]) for the centralized model, and 0.888 (95 % CI: [0.8873, 0.8892]) for the fine-tuned model (cf. Fig 4). The hypothesis that there is no statistical difference between the AUCs was rejected at a significance level of $p = 0.01$. At a threshold of 0.58, the fine-tuned model achieves a sensitivity of 0.87, a specificity of 0.80, and a balanced accuracy of 83% when PAC and PVC are combined into a single anomaly class. However, there are a large number of false positives that are likely caused by the number of normal ECGs of *barely acceptable* quality [52]. The total number of false positives ($n = 721, 185$) with *barely acceptable* signal quality was $424, 337$. By excluding samples with *barely acceptable* signal quality [52], the best-performing approach, FT ResEmb, increased its AUC to 0.91 and demonstrated improvements of 0.0303 in sensitivity, 0.0482 in specificity, and 0.0393 in balanced accuracy.

Examination of the detection performance of the two pathologies shows strong discriminatory power for PVC, with $16, 528$ of $17, 109$ samples correctly identified, corresponding to a sensitivity of 0.9672. For PAC cases, $26, 994$ of $34, 382$ samples were correctly identified, corresponding to a sensitivity of 0.8231. The specificity remained unchanged as the above threshold calculated for all samples was used.

Comparing *Fed ResEmb* to *Cen ResEmb*, the performance loss due to the federated setting was moderate, with reductions of 0.0308 in sensitivity, 0.0676 in specificity, and 0.0493 in balanced accuracy.

**Individual approach**   When determining a per-subject threshold for the fine-tuned model, the AUC value increased to above 0.9 in 14 cases, remained in the range of 0.8 to 0.9 in 4 cases, and was below 0.8 in 2 cases. Subject *p03111* demonstrated notably poor performance, particularly in segments 34 and 49. Upon closer examination and visualization, the signals in these segments exhibited non-smooth slopes and a low dynamic range.

**Table 3. Anomaly detection metrics.**

| Signals | Method | Threshold | Sensitivity | Specificity | Balanced Accuracy |
|---|---|---|---|---|---|
| All | Fed Res | 0.32 | 0.7654 | 0.7405 | 0.7529 |
| All | Fed ResEmb | 0.2603 | 0.7873 | 0.7246 | 0.7559 |
| All | FT Res | 0.1045 | 0.9506 | 0.6813 | 0.816 |
| All | FT ResEmb | 0.5838 | 0.866 | 0.8028 | 0.8344 |
| Excellent | FT Res | 0.0542 | 0.9545 | 0.7391 | 0.8468 |
| Excellent | FT ResEmb | 0.4763 | 0.8963 | 0.851 | 0.8737 |
| All | Cen Res | 0.1266 | 0.8933 | 0.695 | 0.7942 |
| All | Cen ResEmb | 0.4404 | 0.8181 | 0.7922 | 0.8052 |
| p00107 | FT ResEmb | 0.5381 | 0.7143 | 0.8297 | 0.772 |
| p05484 | FT ResEmb | 1.3045 | 0.88 | 0.9071 | 0.8936 |
| p06998 | FT ResEmb | 0.7001 | 0.9182 | 0.8395 | 0.8788 |
| p03984 | FT ResEmb | 1.2415 | 0.9778 | 0.9372 | 0.9575 |
| p03111 | FT ResEmb | 0.0666 | 0.6015 | 0.6758 | 0.6387 |
| p04040 | FT ResEmb | 0.614 | 0.8273 | 0.8292 | 0.8282 |
| p03013 | FT ResEmb | 1.7214 | 0.9293 | 0.9174 | 0.9233 |
| p06607 | FT ResEmb | 0.265 | 0.8276 | 0.7318 | 0.7797 |
| p04219 | FT ResEmb | 0.4177 | 0.8567 | 0.8148 | 0.8358 |
| p08750 | FT ResEmb | 1.9682 | 0.75 | 0.9558 | 0.8529 |
| p05665 | FT ResEmb | 0.9977 | 0.6667 | 0.8734 | 0.77 |
| p09225 | FT ResEmb | 1.4871 | 0.9685 | 0.915 | 0.9418 |
| p08030 | FT ResEmb | 1.8522 | 0.9043 | 0.9612 | 0.9327 |
| p09886 | FT ResEmb | 1.6274 | 0.7838 | 0.9448 | 0.8643 |
| p01851 | FT ResEmb | 1.0777 | 0.9309 | 0.9224 | 0.9266 |
| p01123 | FT ResEmb | 0.0107 | 0.6538 | 0.7444 | 0.6991 |
| p03043 | FT ResEmb | 0.2583 | 0.7778 | 0.698 | 0.7379 |
| p03369 | FT ResEmb | 0.4622 | 0.8688 | 0.7353 | 0.802 |
| p06829 | FT ResEmb | 0.5401 | 0.8481 | 0.8074 | 0.8278 |
| p10969 | FT ResEmb | 1.4471 | 0.9541 | 0.9282 | 0.9412 |

Accumulating the confusion matrices into one yielded $3,078,588$ true-negative, $52,567$ true-positive, $579,094$ false-positive, and $6,893$ false-negative results. These figures correspond to a sensitivity of 0.8841 and a specificity of 0.8417. Detailed subject-related results such as the confusion matrices (S4 Fig), ROC curves (S5 Fig) and the model loss (S6 Fig, S7 Fig, S8 Fig) were included as supplementary material. The reconstruction quality of the federated model is compiled in S9 Fig.

## Inter-subject variations

The embedding space shows a cluster pattern with regard to the subjects (cf. Fig 5). When the embedding space is colored by annotations, no significant clustering can be observed, indicating variations between subjects, e.g., due to differences in the isoelectric baselines, which emphasizes the need for more robust methods when incorporating non-IID data. These findings support the idea that the method requires either subject-invariant anomaly detection using the residuals, personalization by fine-tuning the model and incorporating subject-based metrics such as the embedding space, or both.

## Discussion

### Feasibility

To the best of the authors' knowledge, *Cardioflow* is the first to employ autoencoders in an FL environment on physical smartphone devices to detect heartbeat anomalies in ECG signals

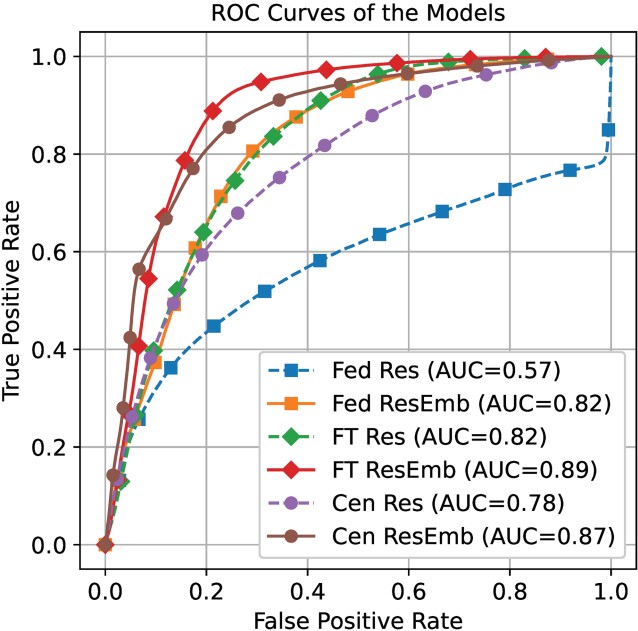

**Fig 4. ROC curve and AUC comparison.**

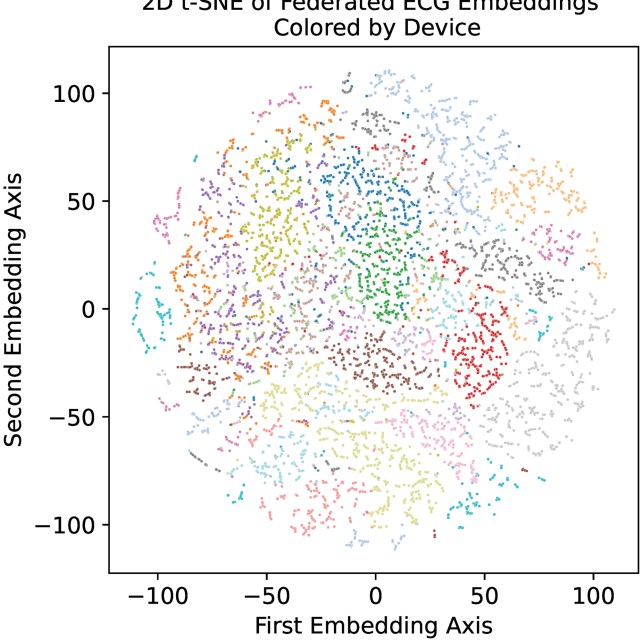

**Fig 5. t-SNE embedding colored by subject (federated model).**

recorded in mobile environments. The study demonstrated the feasibility of such an approach without substantially affecting device functionality on modern hardware.

Although the application is still in the prototype stage, it can be made operational for patient studies with minor changes. The necessary adjustments include: (1) automating the

manually triggered steps (segmentation, embedding, reconstruction) so that they are executed as background tasks using load balancing techniques; (2) improving user feedback mechanisms by replacing technical metrics such as visualizations of the embedding space and residuals in the home view with more user-friendly and better assessable elements; (3) implementing user management and introducing user clusters, similar to the methods used in related research, e.g., *FedCluster* [30].

The application is extendable to any observable pathology and anomaly within the ECG, including previously unknown occurrences and events. The residuals of the autoencoder are expected to be increased for anomalies, ensuring detection beyond normal behavior by design. This characteristic enables robust anomaly detection and may facilitate early detection and intervention in health conditions.

Finally, the application can be customized to monitor other health-related signals such as movement patterns, oxygen saturation, and images, in addition to the ECG. These changes can expand the scope of the application and make the system more versatile for health monitoring applications, with minimal effort required to customize it for different health areas.

The answer to **RQ1** is that federated unsupervised learning in smartphones emulating a real-world environment is computationally feasible. However, the transition from a prototype to a production-ready implementation, particularly for use in clinical trials, needs to be addressed in future work.

## Anomaly detection

The approach showed promising results in predicting anomalous samples from one-second ECG heartbeat segments. In particular, the sensitivity value of 0.88 is medically relevant to avoid missing any positive samples. However, the system tends to categorize normal ECGs as pathological. This tendency can be explained by the unbalanced dataset (cf. S3 Fig), which is common in ECG research, and artifacts in the recorded ECGs that arise from recording in a mobile environment, causing motion noise, such as when standing up or during other physical activities. This noise caused by artifacts affects the system's ability to distinguish between real pathological and normal ECGs. 58.84% of the false-positive signals were classified as signals of *barely acceptable* quality [52], supporting the hypothesis that low-quality signals with a high signal-to-noise ratio impair the analysis. The performance of the centralized approach was slightly better, but did not yield a significant improvement over the results of the federated model.

A follow-up task is to detect movements using the IMU (Inertial Measurement Unit) sensors to identify rapid accelerations that may distort the ECG data and exclude these samples from the analysis. This approach can reduce noise and improve the reliability and trust of the anomaly detection system, which is an important factor in self-supervised environments. The high sensitivity, but also the large number of false positives, motivates to consider the presented system as a tool for filtering the predicted healthy samples and then apply computationally more demanding algorithms to distinguish true anomalies from noisy but healthy samples.

Considering related work on predicting normal, PAC, and PVC beats in an unsupervised setting, e.g., a balanced accuracy of 0.85 based on a random subset of the dataset in *Tan et al. (2021)* [41], the presented unsupervised approach seems to be competitive. Nevertheless, there is a need to explore further mechanisms and adjustments to reduce the number of false positives.

The answer to **RQ2** is therefore that the accuracy of the presented approach for the given pathologies can compete with related work, but with the advantage that it is unsupervised and preserves privacy.

## Inter-subject variations

Fig 5 shows a wide spread over the embedding space, mainly clustered by subjects. The embedding space indicates non-IID data and suggests considering techniques beyond the described FL routine. Therefore, the study fine-tuned the federated model and incorporated individual embedding spaces for subject-invariant anomaly detection.

The results show that fine-tuning the federated model and comparing ECGs between subjects within the embedding space increases the detection accuracy by approximately 0.07 as measured by AUC. In this context, grouping by similar users becomes relevant, as shown in related work, e.g., *FedCluster* [30]. This clustering approach can help to better tune the federated model to the specific characteristics of different user groups, thereby improving the overall performance of the model and the reliability in anomaly detection.

For **RQ3**, on-device fine-tuning facilitates personalization and mitigates inter-subject variation, addressing the challenge of non-IID data. However, future research needs to explore strategies to ensure that fine-tuning does not fundamentally alter the model. The valuable information learned from different subjects could disappear and affect the generalizability and overall performance of the model.

## Limitations

In related literature, more sophisticated autoencoders have been successfully used with high accuracy for detecting anomalies in ECGs, using elements such as LSTM layers, latent vector space regularization, and disentanglement techniques for explainability. However, the functionalities of CoreML limit the use of more complex models that can lead to better and more explainable results.

The iOS ecosystem prohibits calling a function remotely when the application is closed, which makes unobtrusive and randomly scheduled FL as a background task currently infeasible. This limitation requires the predefined minimum number of users to be online and actively participating as a client node.

This study was conducted using twenty mobile devices of varying generations, all operating on the iOS platform. While this provided a cross-device environment, it limits the generalizability of our findings to other operating systems such as Android. The current focus on iPhone devices does not allow for statements on accuracy and performance of *Cardioflow* when simultaneously deployed to different operating systems. Additionally, the relatively small scale of testing devices poses another limitation. In real-world applications, systems typically include a much larger and over the time evolving number of devices, which can lead to complexities such as non-IID data and potentially lower accuracy. However, this larger scale can also contribute to the development of more robust and generalizable models – an aspect not addressed in this study.

Deploying the proposed approach in real-world scenarios also presents several operational challenges. Factors like communication latency and the possibility of clients dropping out due to network instability or power loss can substantially impact system performance. Addressing these issues requires enhanced robustness measures beyond those discussed in Sect 2.3. For example, implementing constraints to ensure that only devices with stable network connections, sufficient power, and minimal conflicting tasks participate in computations can help

mitigate these challenges [42]. However, such constraints must be carefully designed to avoid introducing selection bias.

The approach was tested on the PAC and PVC ECG events and served as a proof of concept. However, there are pathologies with greater relevance and impact, such as atrial fibrillation, which might be a more important topic for investigation. Also, some pathologies may not manifest in individual heartbeats when only a single-lead ECG signal (e.g., Einthoven's lead I [54]) is examined. Therefore, not all potential cardiac diseases can be monitored with mobile devices, such as acute coronary syndrome, myocardial infarction, and left ventricular hypertrophy [4]. The demographic composition of the dataset, particularly the overrepresentation of participants in the later stages of life, present a limitation to the generalizability of the findings. Additionally, the twenty subjects randomly sampled from the large pool of $11,000$ participants may not accurately reflect the overall demographics and may introduce additional bias in the analysis. However, since the dataset did not provide demographic information on an individual subject basis, only general statements can be made about the demographics of the selected samples.

## Future work

Several limitations identified in this study provide a foundation for future efforts: (1) Extending support to additional operating systems; (2) Enabling federated learning routines to run as background processes, thereby enhancing usability without disrupting user activities; (3) Implementing participation constraints based on device capabilities, operational state, and connectivity, as well as exploring client clustering techniques to optimize performance and resource utilization; And (4) repeating the experiment by iteratively sampling different subsets of the Icentia11k dataset and simulate an evolving number of clients over time to demonstrate robustness and generalizability, or conducting studies using a larger set of devices.

The promising results of this study encourage further research using real-time measurements with heart frequency sensors such as the Polar H10 and FL in a more unobtrusive fashion. In this context, automatically excluding low-quality signals is expected to enhance performance, particularly specificity due to a lower count of false positives, as demonstrated in this study when signals of *barely acceptable* quality and lower [52] were omitted. However, it is important to investigate the impact of pathologies on quality measures to ensure that no bias is introduced by systematically excluding pathological ECGs from the analysis. To avoid motion-related distorted ECG readings, it is important to analyze the IMU data simultaneously to exclude segments where sudden movements are detected that can potentially distort the ECG recordings. It is expected that implementing this strategy will reduce type II errors and improve the specificity and precision of the results.

Movement detection is often linked to exercise, which typically results in faster heartbeats and can alter the morphology of one-second ECG segments. This complexity requires the model to effectively capture these variations. By incorporating physical activity data from sources such as smartwatches and vital information like resting heart rate, it becomes possible to either exclude exercise-induced data or develop context-specific models. This enables precision approaches tailored to specific situations, e.g., resting versus exercising. Also, forming client clusters based on characteristics such as average heart rate and demographics can enhance these precision approaches by aligning and comparing participants more closely in their ECG data distributions.

The accuracy of detection could be improved by integrating a few annotated ECGs that serve as a reference point in the embedding space. This semi-supervised approach allows the

expertise of a physician to be incorporated while reducing the burden of ECG screening by evaluating only a few.

Regularizing the latent space, e.g., by using Variational Autoencoders (VAEs) [55] and extensions such as the $\beta$-TCVAE (Total Correlation Variational Autoencoder) proposed by *Chen et al. (2018)* [56], leads to better structured embeddings. Ideally, each dimension of these embeddings corresponds to a specific feature of the ECG, which makes the detection of anomalies in the latent vector space more explainable. Implementing these autoencoder routines requires custom development, as the standard *CoreML* methods do not yet offer sophisticated loss functions beyond mean squared error and cross entropy.

## Conclusion

The research focused on detecting anomalies in mobile ECGs using unsupervised methods while ensuring privacy. A mobile application was developed to record ECG signals and train an autoencoder directly on the device. FL technology was used to merge the decentralized optimized models into a global model. Learning from a large number of individuals has the advantage of producing models with a lower tendency to bias, making them more robust because they are trained on a majority of healthy samples and are not biased by the data of a single subject who may has heart irregularities in most of his ECG recordings.

Running the federated optimization process on twenty mobile devices in a real-world setting proved feasible and yielded high sensitivity and specificity for detecting anomalies (PAC and PVC) in ECGs based on a subset of the Icentia11k dataset. The results are a proof of concept for federated unsupervised learning in a cross-device environment. Future research on clinical validation and deployment in real-life scenarios is needed to further validate the feasibility of the proposed system and clarify open questions, such as its performance with a growing number of participants. The investigation of more complex autoencoder models could also capture a broader range of pathologies.

Overall, this study addressed important topics in mobile ECG diagnostics, including limited annotations, privacy requirements, and non-IID data. Adapting the decentralized approach to other use cases in healthcare could open up additional data sources, such as personal electronic health records on mobile devices. These are widely available but rarely released as open-source datasets. A privacy-preserving, decentralized system that employs unsupervised learning on electronic health records could enable anomaly detection across multiple health domains, particularly benefiting people in regions with limited healthcare access.

## Supporting information

**S1 Fig. Data samples.** Twelve randomly sampled ECG traces to illustrate the data and demonstrate the impact of pathologies on the waveform.
(PNG)

**S2 Fig. Study setting.** Depicts the study setting, including the devices (excluding tablets) and the console input and output of the server.
(PNG)

**S3 Fig. Data Distribution.** Histograms to display the distribution of classes across devices.
(PNG)

**S4 Fig. Confusion Matrices.** Confusion matrices to evaluate the performance of the individual devices.
(PNG)

**S5 Fig. ROC Curves.** ROC curves to assess the performance of the approaches presented for each individual device.
(PNG)

**S6 Fig. Model Loss Federated.** Loss per segment returned per device after training the federated model is presented, with the test segment highlighted to detect possible overfitting.
(PNG)

**S7 Fig. Model Loss Personalized.** Loss per segment for each device after fine-tuning the model is shown, highlighting the test segment to identify possible overfitting.
(PNG)

**S8 Fig. Model Loss Centralized.** Loss per segment returned per device after training the centralized model is presented, highlighting the test segment to identify possible overfitting.
(PNG)

**S9 Fig. Reconstructions.** Reconstructed signals of the federated autoencoder are illustrated by varying individual dimensions of the latent vector space to test the model's ability to generate realistic ECG signals.
(PNG)

## Acknowledgments

The project was supported by the Bavarian State Ministry of Health and Care through the research project DigiMed Bayern. The authors would like to thank Florian Schweizer and Leon Nissen for valuable discussions during the implementation of the mobile application. The authors would also like to thank Daniel Nata Nugraha for providing the relevant information on serializing the weights collected through federated learning.

During the preparation of this work, the authors used ChatGPT (OpenAI, San Francisco, CA, USA) to check the text for readability and to eliminate typos and grammatical errors. Subsequently, the authors reviewed and edited the content as needed and take full responsibility for the content of the publication.

## Author contributions

**Conceptualization:** Maximilian Kapsecker, Stephan M. Jonas.

**Data curation:** Maximilian Kapsecker.

**Formal analysis:** Maximilian Kapsecker.

**Investigation:** Maximilian Kapsecker.

**Methodology:** Maximilian Kapsecker.

**Project administration:** Maximilian Kapsecker.

**Resources:** Maximilian Kapsecker.

**Software:** Maximilian Kapsecker.

**Supervision:** Stephan M. Jonas.

**Validation:** Maximilian Kapsecker.

**Visualization:** Maximilian Kapsecker.

**Writing – original draft:** Maximilian Kapsecker.

**Writing – review & editing:** Maximilian Kapsecker, Stephan M. Jonas.

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
