## [Decision Letter · Decision Letter 0]

14 Nov 2024

PDIG-D-24-00344Cross-device federated unsupervised learning for the detection of anomalies in single-lead electrocardiogram signalsPLOS Digital Health Dear Dr. Kapsecker, Thank you for submitting your manuscript to PLOS Digital Health. After careful consideration, we feel that it has merit but does not fully meet PLOS Digital Health's publication criteria as it currently stands. Therefore, we invite you to submit a revised version of the manuscript that addresses the points raised during the review process. Please submit your revised manuscript within 60 days Jan 13 2025 11:59PM. If you will need more time than this to complete your revisions, please reply to this message or contact the journal office at digitalhealth@plos.org. Please include the following items when submitting your revised manuscript:* A rebuttal letter that responds to each point raised by the editor and reviewer(s). You should upload this letter as a separate file labeled 'Response to Reviewers'. This file does not need to include responses to any formatting updates and technical items listed in the 'Journal Requirements' section below.* A marked-up copy of your manuscript that highlights changes made to the original version. You should upload this as a separate file labeled 'Revised Manuscript with Track Changes'.* An unmarked version of your revised paper without tracked changes. You should upload this as a separate file labeled 'Manuscript'. If you would like to make changes to your financial disclosure, competing interests statement, or data availability statement, please make these updates within the submission form at the time of resubmission. Guidelines for resubmitting your figure files are available below the reviewer comments at the end of this letter. We look forward to receiving your revised manuscript. Kind regards, Martin G FraschSection EditorPLOS Digital Health Martin FraschSection EditorPLOS Digital Health Leo Anthony CeliEditor-in-ChiefPLOS Digital Healthorcid.org/0000-0001-6712-6626 **Journal Requirements:**

1. We ask that a manuscript source file is provided at Revision. Please upload your manuscript file as a .doc, .docx, .rtf or .tex.

2. We do not publish any copyright or trademark symbols that usually accompany proprietary names, eg (R), (C), or TM  (e.g. next to drug or reagent names). Please remove all instances of trademark/copyright symbols throughout the text, including ® on page 17.

3. Please provide separate figure files in .tif or .eps format.

**Additional Editor Comments (if provided):****Reviewers' Comments:** Reviewer's Responses to Questions

**Comments to the Author**

1. Does this manuscript meet PLOS Digital Health’s publication criteria? Is the manuscript technically sound, and do the data support the conclusions? The manuscript must describe methodologically and ethically rigorous research with conclusions that are appropriately drawn based on the data presented.

Reviewer #1: Yes

Reviewer #2: Yes

Reviewer #3: Yes

Reviewer #4: Yes

Reviewer #5: Partly

Reviewer #6: Partly

Reviewer #7: Yes

2. Has the statistical analysis been performed appropriately and rigorously?

Reviewer #1: Yes

Reviewer #2: Yes

Reviewer #3: No

Reviewer #4: No

Reviewer #5: Yes

Reviewer #6: Yes

Reviewer #7: Yes

3. Have the authors made all data underlying the findings in their manuscript fully available (please refer to the Data Availability Statement at the start of the manuscript PDF file)?

Reviewer #1: Yes

Reviewer #2: No

Reviewer #3: Yes

Reviewer #4: Yes

Reviewer #5: Yes

Reviewer #6: Yes

Reviewer #7: Yes

4. Is the manuscript presented in an intelligible fashion and written in standard English?

Reviewer #1: Yes

Reviewer #2: Yes

Reviewer #3: Yes

Reviewer #4: Yes

Reviewer #5: Yes

Reviewer #6: Yes

Reviewer #7: Yes

5. Review Comments to the Author

Reviewer #1: In their manuscript "Cross-device federated unsupervised learning for the detection of anomalies in single-lead electrocardiogram signals", Maximilian Kapsecker and Stephan M. Jonas present the results of a machine learning algorithm trained in a federated manner on 20 mobile phones for detecting anomalies in ECG data.

I think this is interesting work presented in a generally well-written text (with the help of AI tools, as disclosed by the authors) for which the source code has been published. My major points of criticizm are that I do not understand the practical relevance of this training approach and that the analysis of the results leave a few questions, this should be discussed in more detail. This, plus a few minor issues, lead to my recommendation of a major revision.

Training algorithms for analyzing medical data in a federated (privacy-preserving) way and ideally without going through the work of annotating the data is certainly relevant. However, the scenario described here relies on a synchronous availability of federated clients for the initial training, which does not seem very realistic to me, at least not for the specific task here where a large dataset is already available and could be used to provide an initial model that is then fine-tuned locally and individually for each user.

The evaluation of the prediction quality is based on single heartbeats, which is certainly a good basis and likely a deflation rather than an inflation of performance metrics. However, I would expect (I'm not an expert in this field, though) that one is actually interested in a per-patient prediction of premature contractions (or per-patient on a given day). A prediction quality assessment on such aggregated data could result in higher values for appropriate metrics if mis-classifications even out over multiple heartbeats.

Given that the present evaluation is based on a dataset distributed to the federated clients, a comparison to predictive performance after centralized training would be possible—and interesting to assess whether federation has a negative impact in this application.

If I understand correctly, the authors used 20 of 11,000 cases from a publicly available dataset. In the present scenario, it makes perfect sense to use one per client, and should not be viewed as using very little of the available data. However, this opens a good opportunity to assess how well the approach generalizes to other data by repeatedly sampling other 20 cases and obtaining confidence intervals for an overall performance metric. Also, it would be interesting to see how deterministic the results are for one set of 20 cases, i.e., to which extent performance differs between repeated runs even if the order of data used, accessible random seeds, etc. are fixed. (I'm more familiar with training deep neural networks on GPUs where this can be an issue; I'm not sure how deterministic the combination of federated training, mobile phones, the autoencoder architecture and the training scheme used here can be expected to be.) Combined with a federated-vs-centralized assessment, a centralized training on more than 20 clients could be used to estimate generalization in case more federated clients were involved.

The limitations section briefly touches on using the presented approach in a real-world scenario. I would recommend elaboating on this a bit, as there will be further complicating factors like communication latency, clients dropping out (loss of network connection, loss of power), and others, which may introduce robustness requirements in addition to the requirements mentioned in Section 2.3.

Detailed comments

l 2: "mobile devices have evolved" is quite vague, is this about mobile phones having evolved to tools for multiple purposes, measurement devices having evolved to analysis capabilities?

l 56: I think "respectively" should be followed by a comma (but I'm not a native speaker)

l 112: the dataset itself should be cited here in addition to publications about the dataset

l 113: I would find "3 to 14 days" easier to read.

l 121: an explanation of "fifty segments" in terms of number of samples or time span covered would be helpful

l 240f: I would write the sentence as "The T-distributed stochastic neighbor (t-SNE) method was used …"

Fig. 4: I would consider omitting this figure, it is not particularly helpful. The message of "20 clients" could instead be included in Fig 3, naming the clients 1, 2, 3, 20 and adding "…"

Fig. 5: Similarly, this screenshot of console output does not contain too many insights relative to the size of the figure. Maybe make this one a supplementary figure?

l 250: should read "Euclidean" and not start a new paragraph after the formula

Fig. 7: Here, I would recommend using different line types to make the figure greyscale-/colorblind-safe. (Same problem in Fig 6, but I doubt using different symbols would really help there.)

l 453: should read "proved feasible"

l 456: I think "proof of concept" should not be hyphenated here

Captions for supporting figures: I would rephrase the captions to omit "The figure shows"

l 481f: Unless the template text needs to be kept verbatim, I would use the applicable variant: "the authors … take …"

References: Should be double-checked for punctuation in URLs and arXiv identifiers, as well as capitalization of names in titles (e.g., Bayes) and journal names.

Figure S2: The text is very small and hard to read.

Figure S6: The variation is described only vaguely in the caption and in the text. I would appreciate some more detail on the range of variation.

Reviewer #2: The manuscript presents a framework for the analysis of ECG signals from wearable devices, following a federated learning (FL) approach. The goal of the authors is to test the deployment of a machine learning autoencoder model within a GUI on mobile devices (iOS namely) which allow for analysis and semi-clinical output. The approach starts with a base model trained on open-source data and if self-supervised tuned on individual readings. The authors have a clear goal, nonetheless, their narrative has some gaps which need to be addressed before publication.

Also, I believe the focus of the manuscript should be “a model which is deployed as part of an app” but not focus as much the app per say – focus on the technology and not the vessel of it.

Because of the inherent high value of the research, the manuscript should be classified as “Accepted with major revisions.”

Suggestion/needed additions are as follow per section:

Author Summary

At the author summary, the authors mention:

“mitigate the impact of data heterogeneity, we implemented on-device fine-tuning of the federated model for personalization”

The section 2.5 titled “Evaluation” should have explain this method better and yet it did not. It is not clear how ground-truth data for validation is obtained.

Introduction:

Grammar suggestions:

Line 99, change question to: How feasible is autoencoders implementation for federated unsupervised learning, in a cross-device real-world scenario?

Line 101 correct to: What is the performance of autoencoders in the detection of electrocardiogram anomalies as part of a cross-device federated unsupervised environment?

Line 103 correct to: Can on-device fine-tuning be sufficient for personalization and detecting baseline changes, thus addressing the challenge of non-IID data

Materials and Methods

Regarding data, the use of open source dataset Icentia11k is clear. Nonetheless, if all the dataset was used for training or not, is unclear. Further, the data loaded into the mobile apps is unclear regarding its origin – are these datapoints also part of the open source dataset, or were these newly generated by users? If the later is true, is there an ethic approval?

The authors also miss the explanation of why they target development into the iOS platform and used the Polar H10 as measurement device (again, it misleading as while a measuring device is listed, no human-lead experiments are mentioned). Several wearables, of different brand provide ECG recording, analysis and export. Thus, the sole deployment on apple devices and one sole type of wearable seems short at best. Thus, the authors are asked to either run experiments for the android ecosystem and also use different wearables or given better explanations regarding their decisions (materials and methods section).

Thus, the authors must at least provide:

• Explanations regarding the sole targeting of iOS

• Explanations of the sole use of Polar H10 device

• Cohort of human testers (particularly important since the open-source dataset is centered on third-age population)

It is missing how the autoencoder was trained, namely hardware. Is it also missing how the model was adapted to run in CoreML.

Regarding the federated learning pipeline, little is mention about tis implementation and if each individual refined model (in their respective devices) is then fetch back to the central server for model merging (whole purpose of federated learning – gathering data within the models without having access to the data per say).

Results:

In section 3.1. the authors miss on mention what Modern UI libraries were used for visualization and why.

There should be a section for results on the base model itself. Since you took an open source dataset, and used a costume autoencoder, then, validation metrics should have been mentioned and not only metrics after deployment in the devices. A common 70-30% training-validation data split is suggested.

The base model can then be compared to the output after fine tunning. There should also be a comprehensive description of the fine-tune results (as text, not only figures).

The models acronyms need better explaining as well.

Reviewer #3: I have reviewed the manuscript titled "Cross-device Federated Unsupervised Learning for the Detection of Anomalies in Single-lead Electrocardiogram Signals." The authors present a novel idea of heartbeat anomaly detection using auto-encoders in a real-world scenario involving 20 mobile phones.

Strengths:

1- The manuscript addresses a critical aspect of current clinical problems in cardiology, highlighting its significant scientific value.

2- The proposed method of using federated unsupervised learning for anomaly detection in ECG signals is innovative and demonstrates practical applicability.

3- Conducting experiments on 20 mobile phones showcases the feasibility of the approach in real-world scenarios, which enhances the practical relevance of the study.

Suggestions for Improvement:

1- Effect of Number of Clients: It would be interesting to see the effect of the number of clients on the final result. I suggest the authors run the same experiments with different numbers of clients (e.g., 4, 8, 16, 20) and report how the model's performance changes. This would provide a deeper understanding of the scalability and robustness of the proposed method.

2- Comparison with Centralized and Local Learning: As the data is publicly available, it would be valuable to compare the results of centralized learning (all data in one place, without federated learning) and local learning (each client is trained and evaluated based on its data) against the federated learning approach already included in the study. This comparison would offer insights into the relative advantages and potential limitations of each method.

Overall, the manuscript presents a significant contribution to the field of ECG signal processing and federated learning. The study is scientifically important and practically relevant. Incorporating the suggested experiments and comparisons would further strengthen the manuscript and provide a more comprehensive evaluation of the proposed approach.

Reviewer #4: The authors proposed to use reconstruction errors and embeded vectors from trained autoencoders using federated learning on mobile phones for anomaly detection of ECG singals collected by consumer device, Polar H10 on open access dataset. They introduced also on-device fine-tuning for personalization to address the challenge posed by non-IID data. The proof-of-concept type of study seems to be promising but I think there are a few things which the authors need to address before the manuscript shall be accepted for publication.

1) The authors mentioned that "58.84% of the false-positive signals were classified as signals of medium quality". If I am not mistaken, the signal quality assessment was done via the SQI score proposed by reference 48. The authors did not mention the quantitative definition of what was considered as "medium quality". It would be interesting to see the results trained and tested on the "high quality" data excluding those "medium equality" ones.

2) From the ROC curves shown in Figure 7, the curve for FT ResEmb seemed very likely to be significantly better than the others, but the authors did not performed any statistical tests to verify it. It will be better if the authors can compute confidence intervals to the ROC curves and perform DeLong tests on the results.

3) For the limitation of the study, the authors did not mention that all training and testing were restricted to iphone devices. Android phones have a dominant market share globally. It can be a serious limitation for applications.

4) The results on p03111 were overall worse than the others. It would be interesting to look into it and discuss what caused this difference.

Reviewer #5: The authors conducted a comprehensive study on Federated Learning (FL) for ECG signal anomaly detection. They developed an iOS app to collect ECG data, train a model locally, and share model weights with a central server for global updates. The model was evaluated on unseen data from the same 20 users who participated in training, achieving a sensitivity of 0.88 and a specificity of 0.84. Several aspects require clarification:

- How was the decision made regarding the latent vector size in the autoencoder?

- Given the sigmoid output activation, input signals must be normalized. Please elaborate on the normalization method used.

- How would the model's performance evolve with a large number of users? Would it be continuously updated, and how would this impact the accuracy for early users?

- Is there a risk of the model learning to reconstruct anomalies if it encounters a sufficient number of anomalies in the training dataset?

- Could you clarify the process for determining individual anomaly thresholds?

- How does the algorithm handle faster heart rates, such as those during exercise, given that 1-second signals are extracted?

- Why were annotations transferred to devices, while they should be needed only for evaluation of the results?

- Using the first segments for testing might introduce bias due to learning effects or device adjustments.

- Please clarify the meaning of "unknown beats" and whether they were excluded from the study. Could normal and anomalous beats be inferred from rhythm annotations?

- Could you define the embedding vector and embedding space used in the autoencoder?

- It would be nice to analyze the results on how increasing the number of users can affect the results. The model can be saved and used for another set of 20 users, right?

Reviewer #6: Overall Evaluation:

The authors present an interesting proof-of-concept study exploring the use of federated learning with autoencoder models for detecting cardiovascular anomalies from single-lead ECG data in a mobile environment. The study shows promise for potential applications in mobile health, especially for consumer health, where it could help detect ECG anomalies via mobile-phone-based applications. However, there are several key areas where the study's generalizability, methodology, and validation need further clarification or improvement. These issues are outlined below and should be addressed to strengthen the manuscript.

1. Generalizability of the Model:

The dataset used to train the model appears to come from a population with an average age of 62 years, according to the source referenced (https://physionet.org/content/icentia11k-continuous-ecg/1.0/). There is no discussion of the diversity of the dataset, particularly in terms of race and sex, which is crucial for understanding the model's performance across various demographic groups.

As the model’s performance is highly dependent on the training data, the lack of demographic diversity may limit its generalizability. The authors should clarify this in the manuscript and, if applicable, acknowledge this as a limitation.

2. Inter-Subject Variability:

The authors acknowledge significant inter-subject variability in their analysis, as indicated by distinct clusters in the embedding space based on individual subjects. This suggests that ECG signals vary significantly across individuals, potentially impacting the model's sensitivity and accuracy when applied to populations outside the demographic represented in the training data.

The manuscript should either address how the model handles this variability or clearly state this as a limitation. Additionally, the demographic distribution of the users of the 20 mobile devices tested should be disclosed to assess the impact of these differences on the results.

3. Limited Device Testing:

The proof-of-concept was only tested on iPhone users, which raises concerns about the generalizability of the tool to users of other devices or operating systems. This should be addressed by either expanding testing to a broader range of devices or including this limitation in the manuscript.

4. Lack of Annotation Clustering:

In Section 3.2, the authors note that no significant clustering was observed when the embedding space was colored by annotations (e.g., normal, PAC, PVC), indicating that inter-subject differences overshadow the variations between different types of heartbeats within individuals.

This raises concerns about the tool's ability to detect anomalies within a single subject’s ECG. While the authors suggest that fine-tuning the model to individual subjects could improve accuracy, they need to demonstrate this with data, especially for subjects who do not fit within the distribution of the training data. If not, this should be listed as a limitation of the approach.

5. High False Positive Rate:

The authors should explicitly mention the high false positive (FP) rate observed during testing as a limitation. Additionally, strategies to mitigate this issue, or adjustments to the tool’s intended goal to account for this, should be discussed.

6. Lack of External Validation:

The model was developed and tested on a specific dataset, with no evidence provided for external validation. The lack of external validation is a critical concern for a model intended for real-world deployment. The authors should either provide evidence of external validation or clearly state this as a limitation of the study.

The study provides an innovative approach to ECG anomaly detection in a mobile setting. However, addressing the concerns related to generalizability, inter-subject variability, false positives, and external validation is crucial for the broader applicability of the tool. The authors should either address these issues or acknowledge them as limitations in the manuscript.

I look forward to the revision.

Reviewer #7: Dear authors, thank you for your very interesting manuscript. Overall the paper reads very well. However, I would suggest some improvements, especially in terms of readability, but also to clarify the study's test status.

Introduction

- The introduction is a little overwhelming to read if you want to get a quick overview of the essence of the work. I would suggest starting with a short introduction that quickly leads to the objective and research questions (both of which I would assume are at the end of the introduction), and move the literature review and descriptions of related work to another subsection.

- You define three research questions and address them in the paper. Since you added them, I would assume you give a summary per question in the discussion: Were each of these questions answered successfully? You could also organise the results more clearly based on these questions to make them easier to follow.

- Table 1 somehow looks like a skewed picture. Is it just the font used?

Methods

- All figures: The figures have a title, but usually no description. Normally, an figure should be self-explanatory through its title and labelling.

- P6, L153: Something with the wordings seems to be wrong here.

- P6, L159: Maybe its better to reference the git repository here instead to the current location

- Figure 3: You are showing a lot of screenshots. Please add some descriptions of each to the caption.

- Section 2.3: I would strongly suggest to use a consistent wording for watch and smartphone. I think all FR are targeted at “your” phone-App, but maybe you also have FR for the watch itself or the watch-App on the phone?

- Section 2.3: Since this is not a development white paper, it is not really of interest that you successfully implemented the FR you defined beforehand. I would suggest to just write the features you implemented.

- Section 2.4: Is not the whole paper the study? This section might better be titled with “technical setting“

- Figure 4 and 5 seem not to be so important. Figure 4 is similar to Figure 3 and could probably be removed or moved to the online supplement

- L207ff: Did you use the FL customization functions for this study? Is this really useful instead of a common-server-provided configuration?

- In the ‘Description of the app’ section, some details of the actual calculation are described that could/should rather be presented in the ‘Study’ or ‘Evaluation’ section. For example, the fact that the snippets are cut by 1 second is described under ‘Settings “View”’, which is not directly understandable, since it is also relevant for the analysis.

Results

- In the methods you write that the data comes from Physionet. In L276ff you say that they were extracted from the 20 devices. Could you clarify this? If they come from the devices: on which subjects were they measured? Would an informed consent be necessary?

- L293ff: The restriction should perhaps be moved to the discussion.

- Section 3.3: You take up the method description here. But you are actually describing the evaluation scenario again, which should somehow be defined in the Evaluation section. I would suggest defining a specific table in the Method Evaluation section that describes the actual models you used for the whole evaluation (i.e. as a specification for Table 2). The whole methodology should be understandable without reading the results. The results could be written in a more condensed way.

6. PLOS authors have the option to publish the peer review history of their article (what does this mean?). If published, this will include your full peer review and any attached files.

**Do you want your identity to be public for this peer review?** For information about this choice, including consent withdrawal, please see our Privacy Policy.

Reviewer #1: No

Reviewer #2: No

Reviewer #3: No

Reviewer #4: **Yes: **Lei Xu

Reviewer #5: No

Reviewer #6: No

Reviewer #7: No

---

## [Decision Letter · Decision Letter 1]

6 Feb 2025

PDIG-D-24-00344R1Cross-device federated unsupervised learning for the detection of anomalies in single-lead electrocardiogram signalsPLOS Digital Health Dear Dr. Kapsecker, Thank you for submitting your manuscript to PLOS Digital Health. After careful consideration, we feel that it has merit but does not fully meet PLOS Digital Health's publication criteria as it currently stands. Therefore, we invite you to submit a revised version of the manuscript that addresses the points raised during the review process. Please submit your revised manuscript within 30 days Mar 08 2025 11:59PM. If you will need more time than this to complete your revisions, please reply to this message or contact the journal office at digitalhealth@plos.org. Please include the following items when submitting your revised manuscript:* A rebuttal letter that responds to each point raised by the editor and reviewer(s). You should upload this letter as a separate file labeled 'Response to Reviewers'. This file does not need to include responses to any formatting updates and technical items listed in the 'Journal Requirements' section below.* A marked-up copy of your manuscript that highlights changes made to the original version. You should upload this as a separate file labeled 'Revised Manuscript with Track Changes'.* An unmarked version of your revised paper without tracked changes. You should upload this as a separate file labeled 'Manuscript'. If you would like to make changes to your financial disclosure, competing interests statement, or data availability statement, please make these updates within the submission form at the time of resubmission. Guidelines for resubmitting your figure files are available below the reviewer comments at the end of this letter. We look forward to receiving your revised manuscript. Kind regards, Krasimira Tsaneva-AtanasovaAcademic EditorPLOS Digital Health Krasimira Tsaneva-AtanasovaAcademic EditorPLOS Digital Health Leo Anthony CeliEditor-in-ChiefPLOS Digital Healthorcid.org/0000-0001-6712-6626   **Additional Editor Comments (if provided):****Reviewers' Comments:** Reviewer's Responses to Questions

**Comments to the Author**

1. If the authors have adequately addressed your comments raised in a previous round of review and you feel that this manuscript is now acceptable for publication, you may indicate that here to bypass the “Comments to the Author” section, enter your conflict of interest statement in the “Confidential to Editor” section, and submit your "Accept" recommendation.

Reviewer #1: (No Response)

Reviewer #4: All comments have been addressed

Reviewer #6: All comments have been addressed

Reviewer #7: All comments have been addressed

Reviewer #8: All comments have been addressed

2. Does this manuscript meet PLOS Digital Health’s publication criteria? Is the manuscript technically sound, and do the data support the conclusions? The manuscript must describe methodologically and ethically rigorous research with conclusions that are appropriately drawn based on the data presented.

Reviewer #1: Yes

Reviewer #4: Yes

Reviewer #6: Yes

Reviewer #7: Yes

Reviewer #8: Yes

3. Has the statistical analysis been performed appropriately and rigorously?

Reviewer #1: Yes

Reviewer #4: Yes

Reviewer #6: Yes

Reviewer #7: Yes

Reviewer #8: Yes

4. Have the authors made all data underlying the findings in their manuscript fully available (please refer to the Data Availability Statement at the start of the manuscript PDF file)?

Reviewer #1: Yes

Reviewer #4: Yes

Reviewer #6: Yes

Reviewer #7: Yes

Reviewer #8: Yes

5. Is the manuscript presented in an intelligible fashion and written in standard English?

Reviewer #1: Yes

Reviewer #4: Yes

Reviewer #6: Yes

Reviewer #7: Yes

Reviewer #8: Yes

6. Review Comments to the Author

Reviewer #1: The authors have implemented my suggestions for the manuscript and the supporting information in a convincing way. I recommend to accept the manuscript after a minor revision (referring to line numbers in the "clean" version of the revised manuscript):

l 217: I would suggest framing this slightly differently to distinguish what is necessary in the present prototype from how it would be used in later routine, along the lines of "in the present implementation, ECG data is uploaded as a CSV file ..."

l 260 should read "in Section 2.3" (or actually reference the respective subsection by title, I don't think PLOS Digital Health uses numbered sections, please double-check)

l 334 and possibly later locations where signal quality is mentioned: I would suggest repeating "[51]" to avoid the impression that this is some ad-hoc assessment. The classification is described as part of the Methods, but rather hidden in the text (which I think is fine).

l 422 and 515: I would suggest replacing "significant" with "substantial" in contexts not describing statistical significance to avoid potential confusion.

l 503: I would suggest phrasing the limitation on iOS devices slightly differently, along the lines of "our current focus on iPhones", in order to avoid suggesting the approach is generally impossible on Android or other platforms.

l 514: I would suggest adding a literature reference after "Goldberger lead I" for readers unfamiliar with this term.

l 544: I would suggest rephrasing to "with heart frequency sensors such as the Polar H10".

Reviewer #4: The finding of non-smoothness of signals might explain the lower performance of the model on subject p03111 is interesting. For non-expert readers about ECGs like me, it might be good to show a few typical cases of how pathologically relevant abnormal ECG signals should look like in the supplementary material. Other than that, I don't have more concerns about the updated manuscript.

Reviewer #6: the authors have addressed all my concerns.

Reviewer #7: Dear authors. Thank you for the good revision. All my comments were well addressed and I do not have any further comments.

Reviewer #8: The manuscript effectively introduces the novel application of federated unsupervised learning for anomaly detection in single-lead ECG signals, leveraging decentralized data on consumer devices. The background provides a comprehensive overview of the challenges and potential of this approach. However, it could benefit from a deeper exploration of the limitations in existing methods for ECG anomaly detection, particularly in unsupervised contexts. This would better situate the study's contributions within the broader research landscape.

The methodological framework is robust, detailing the use of a linear autoencoder model within a federated learning pipeline. The integration into a mobile application is a significant strength, demonstrating real-world applicability. However, more explicit details on ECG data preprocessing, such as segmentation and normalization, would enhance clarity. Additionally, the demographic skew in the Icentia11k dataset should be discussed to address potential biases.

The results section is thorough, with impressive sensitivity and specificity metrics. However, the discussion should address the high false positive rate, particularly in mobile health applications where user trust is critical. An analysis of factors contributing to false positives, such as signal noise, would provide valuable insights for improvement.

The conclusion succinctly summarizes the study's contributions, highlighting federated unsupervised learning's potential in mobile health. However, it would benefit from explicit articulation of broader implications for mobile health and anomaly detection. A forward-looking perspective on future research directions, including clinical validation and real-world deployment, would strengthen the conclusion.

7. PLOS authors have the option to publish the peer review history of their article (what does this mean?). If published, this will include your full peer review and any attached files.

**Do you want your identity to be public for this peer review?** For information about this choice, including consent withdrawal, please see our Privacy Policy.

Reviewer #1: No

Reviewer #4: **Yes: **Lei Xu

Reviewer #6: **Yes: **Pushkala Jayaraman

Reviewer #7: No

Reviewer #8: **Yes: **Abhik Choudhury

---

## [Editor Report · Decision Letter 2]

17 Feb 2025

PDIG-D-24-00344R2Cross-device federated unsupervised learning for the detection of anomalies in single-lead electrocardiogram signalsPLOS Digital Health Dear Dr. Kapsecker, Thank you for submitting your manuscript to PLOS Digital Health. After careful consideration, we feel that it has merit but does not fully meet PLOS Digital Health's publication criteria as it currently stands. Therefore, we invite you to submit a revised version of the manuscript that addresses the points raised during the review process. Please submit your revised manuscript within 30 days Mar 19 2025 11:59PM. If you will need more time than this to complete your revisions, please reply to this message or contact the journal office at digitalhealth@plos.org. Please include the following items when submitting your revised manuscript:* A rebuttal letter that responds to each point raised by the editor and reviewer(s). You should upload this letter as a separate file labeled 'Response to Reviewers'. This file does not need to include responses to any formatting updates and technical items listed in the 'Journal Requirements' section below.* A marked-up copy of your manuscript that highlights changes made to the original version. You should upload this as a separate file labeled 'Revised Manuscript with Track Changes'.* An unmarked version of your revised paper without tracked changes. You should upload this as a separate file labeled 'Manuscript'. If you would like to make changes to your financial disclosure, competing interests statement, or data availability statement, please make these updates within the submission form at the time of resubmission. Guidelines for resubmitting your figure files are available below the reviewer comments at the end of this letter. We look forward to receiving your revised manuscript. Kind regards, Krasimira Tsaneva-AtanasovaAcademic EditorPLOS Digital Health Krasimira Tsaneva-AtanasovaAcademic EditorPLOS Digital Health Leo Anthony CeliEditor-in-ChiefPLOS Digital Healthorcid.org/0000-0001-6712-6626   **Additional Editor Comments (if provided):****Reviewers' Comments:**   **Figure resubmission:** While revising your submission, please upload your figure files to the Preflight Analysis and Conversion Engine (PACE) digital diagnostic tool, https://pacev2.apexcovantage.com/. PACE helps ensure that figures meet PLOS requirements. To use PACE, you must first register as a user. Registration is free. Then, login and navigate to the UPLOAD tab, where you will find detailed instructions on how to use the tool. If you encounter any issues or have any questions when using PACE, please email PLOS at figures@plos.org. Please note that Supporting Information files do not need this step. If there are other versions of figure files still present in your submission file inventory at resubmission, please replace them with the PACE-processed versions. **Reproducibility:** To enhance the reproducibility of your results, we recommend that authors of applicable studies deposit laboratory protocols in protocols.io, where a protocol can be assigned its own identifier (DOI) such that it can be cited independently in the future. Additionally, PLOS ONE offers an option to publish peer-reviewed clinical study protocols. Read more information on sharing protocols at https://plos.org/protocols?utm_medium=editorial-email&utm_source=authorletters&utm_campaign=protocols

---

## [Editor Report · Decision Letter 3]

19 Feb 2025

Cross-device federated unsupervised learning for the detection of anomalies in single-lead electrocardiogram signals

PDIG-D-24-00344R3

Dear Mr. Kapsecker,

We are pleased to inform you that your manuscript 'Cross-device federated unsupervised learning for the detection of anomalies in single-lead electrocardiogram signals' has been provisionally accepted for publication in PLOS Digital Health.

Best regards,

Krasimira Tsaneva-Atanasova

Academic Editor

PLOS Digital Health